# Immunometabolic Regulation of Vaccine-Induced Antibody Responses in Aging Mice

**DOI:** 10.3390/vaccines12090960

**Published:** 2024-08-26

**Authors:** Daniela Frasca, Maria Romero, Laura Padula, Eva Fisher, Natasa Strbo

**Affiliations:** 1Department of Microbiology and Immunology, University of Miami Miller School of Medicine, Miami, FL 33136, USA; mromero5@med.miami.edu (M.R.); lpadula@med.miami.edu (L.P.); efisher@med.miami.edu (E.F.); 2Sylvester Comprehensive Cancer Center, University of Miami Miller School of Medicine, Miami, FL 33136, USA

**Keywords:** B cells, immunometabolism, inflammation, vaccine

## Abstract

Immune cells undergo metabolic reprogramming to meet the demands associated with immune responses. The effects of aging on these pathways and on the metabolic phenotype of the immune cells participating in antibody responses to vaccines are still largely unknown. Here we used a vaccine for SARS-CoV-2 that utilizes the cellular heat shock chaperone glycoprotein 96 (gp96), engineered to co-express SARS-CoV-2 Spike (spike) protein (gp96-Ig-S). Results show that this vaccine induces comparable B cell primary responses in young and old mice at later time points, but a significantly lesser secondary response in old as compared to young mice, with the antibodies generated in the secondary response being also of lower avidity. This occurs because aging changes the B cell metabolic phenotype and induces hyper-metabolic B cells that are associated with higher intrinsic inflammation and decreased protective antibody responses. However, the gp96-Ig-S vaccine was found to be effective in significantly reducing the metabolic/inflammatory status of B cells from old mice, suggesting the possibility that targeting metabolic pathways may improve immune function in old mice that do not respond adequately to the vaccine.

## 1. Introduction

The antibody response to infections and vaccines decreases with advancing age in both mice and humans, and is characterized by a decreased secretion of protective antibodies and an increased secretion of autoimmune antibodies, which are pathogenic [1]. Our previously published work on humans and mice has shown deficiencies in B cell function, mainly due to the increased chronic low-grade inflammation with age, inflammaging [2]. Inflammaging is linked to several metabolic abnormalities, such as dysregulated nutrient uptake [3], decreased insulin sensitivity [4,5], and dysfunctional mitochondria [6,7]. These abnormalities not only provide energy to support anabolic processes that lead to the secretion of inflammatory products that fuel inflammaging [8], but also lead to the release into the blood of metabolites, some of which (such as Fatty Acids (FAs), long-chain acylcarnitines, and lactate) have been positively associated with inflammaging [9,10,11,12] and immune cell dysfunction [13,14,15], suggesting the importance of studying immunometabolism to fully understand the pathways of immune cell activation and function.

Immune cells undergo metabolic reprogramming to meet the bioenergetic and biosynthetic demands needed for optimal immune responses, and rely on anaerobic glycolysis and oxidative phosphorylation, two processes during which glucose-derived pyruvate generates lactate in the cytosol and acetyl-CoA in the mitochondria, respectively. The role of the overall metabolic status, as well as that of the metabolic pathways/markers of immune cells, and their implications for antibody responses to vaccines, is largely unknown. Also, the effects of aging on these pathways/markers are still unknown.

In this study, we have measured the quantity and quality of humoral responses in young and old mice vaccinated with a vaccine for SARS-CoV-2. This vaccine utilizes the cellular heat shock chaperone glycoprotein 96 (gp96), which is a safe and effective tool for delivering antigens to the immune system; it is a genetically engineered construct of gp96 with the C-terminal KDEL-retention sequence replaced with the Fc portion of an IgG1. The gp96-Ig chaperones proteins/peptides to the immune system and induces potent specific innate and adaptive immune responses, as shown not only for pathogen-derived antigens but also for tumor-derived antigens, as reviewed in [16]. Our recently published results with gp96 engineered to co-express SARS-CoV-2 S (spike) protein (gp96-Ig-S) have indeed shown strong CD4+ and CD8+ T cell responses [17], as well as B cell/antibody responses [18], induced by the gp96-Ig-S vaccine in young mice. The gp96-peptide complexes, either released from target cells (damaged or infected) or administered via vaccination (gp96-Ig), are internalized by antigen-presenting cells, and in the context of MHC-I and MHC-II, prime antigen-specific CD8+ and CD4+ T cells, respectively [19,20,21].

Our results herein show that the gp96-Ig-S vaccine induces comparable B cell primary responses in young and old mice at later time points, but a significantly lesser secondary response in old as compared to young mice, with the antibodies generated in the secondary response being also of lower avidity. This occurs because aging changes the B cell metabolic phenotype and induces hyper-metabolic B cells that are associated with higher intrinsic inflammation and decreased protective antibody responses. However, the gp96-Ig-S vaccine was able to significantly reduce the metabolic/inflammatory status of B cells from old mice, suggesting that it is possible to target metabolic pathways and improve immune function in old mice that do not respond adequately to the vaccine.

## 2. Materials and Methods

### 2.1. Mice

Young and old C57BL/6 male mice were obtained from the National Institutes on Aging. The mice were 3–4 months old (young) and 18–22 months old (old). The mice were maintained in our facility, which is AAALAC-certified. The mice were acclimated for a minimum of 7 days before vaccination. We excluded from the experiments any mice with tumors and infections (skin and eye). All studies were IACUC approved (protocol #19-199-LF and #21-069).

### 2.2. Vaccine

The generation of vaccine cell lines and the characterization of vaccine efficacy have already been described [17]. Male C57BL/6 mice were injected intraperitoneally with gp96-Ig-S vaccine cells (equivalent numbers of vaccine cells producing 1 µg/mL gp96-Ig-S) at t0 and at t28 (to evaluate secondary responses). The mice were sacrificed at t35.

### 2.3. Flow Cytometry

The composition of the splenic B cell pool was evaluated as follows. Spleens were isolated and single-cell suspensions were obtained. The cells were then stained for 20 min at room temperature, using the following mix of reagents and antibodies to measure the three major splenic B cell subsets, follicular (FO), marginal zone (MZ), and age-associated B cells (ABCs): Live/Dead reagent (ThermoFisher, Waltham, MA, USA), anti-CD45 (Biolegend, San Diego, CA, USA 103130), anti-CD19 (BD 557655), anti-CD93 (AA4.1, eBioscience 25-5892-81), anti-CD43 (BD 560663), anti-CD21/CD35 (BD 553818) and anti-CD23 (BD 553139). The B cell subsets, gated on Live/CD45+ CD19+, were as follows: FO B cells (CD23+ CD21−), MZ B cells (CD23− CD21+), and ABCs (CD23− CD21−). Anti-CD93 and anti-CD43 antibodies were used to exclude transitional and B1 B cells, respectively. After membrane staining, the cells were fixed with BD Cytofix (BD 554655). Results were acquired on an LSR-Fortessa (BD) instrument. The results were analyzed using FlowJo 10.5.3 software.

### 2.4. B Cell Sorting

We used CD19 MicroBeads (Miltenyi Biotec, Auburn, CA, USA 130-121-301) to positively isolate splenic B cells, following the MiniMACS protocol: 20 µL Microbeads + 80 µL PBS, for 10^7^ cells, 20 min incubation at 4 °C. The cells were evaluated by flow cytometry for purity and were found to be 90–95% CD19+. Then, the B cells were divided into 2 aliquots: one was used for Seahorse experiments and the other was resuspended in lysis buffer for mRNA isolation using the µMACS mRNA isolation kit (Miltenyi Biotec).

The subsets of FO and ABC B cell subsets were sorted by a Sony SH800 cell sorter using the markers above.

### 2.5. mRNA Extraction and Quantitative (q)PCR

The mRNA was extracted from identical numbers of unstimulated total B cells from the young and old mice, or from identical numbers of unstimulated FO and ABCs from the young and old mice, using the µMACS mRNA isolation kit (Miltenyi Biotec), following the manufacturer’s protocol. The mRNA was eluted into 75 µL of preheated elution buffer and stored at −80 °C until use.

Reverse transcriptase (RT) reactions were performed in a Mastercycler Eppendorf Thermocycler to obtain cDNA as follows: 40 min at 42 °C and 5 min at 65 °C. The cDNA was amplified in MicroAmp 96-well plates, and then run in the ABI 7500 machine. Calculations were made by determining the cycle number at which transcripts reached a significant threshold (Ct). The value of the amount of the target gene, relative to GAPDH, was calculated and expressed as ΔCt. The results show qPCR values, which are measures of the RNA expression of the target genes, relative to the housekeeping gene GAPDH, calculated as 2^−ΔCts^. The reagents and primers for qPCR amplification, all from ThermoFisher, were as follows: *LDHA*, Mm01612132_g1; *PDHX*, Mm00558275_m1; *TNF*, Mm00443258_m1; *IL-6*, Mm00446190_m1; *p16^INK4^*, Mm00494449_m1; *p21^CIP1/WAF1^*, Mm00432448_m1.

### 2.6. Enzyme-Linked Immunosorbent Assay (ELISA)

To measure gp96-Ig-S vaccine-specific IgG antibodies, ELISA plates were coated with recombinant NCP-CoV (2019-nCoV) Spike protein (S1 + S2 ECD). Serum samples were serially diluted, and data were analyzed using a sigmoidal curve model or parallel line analysis after log transformation. The detection antibody was a biotin-conjugated goat anti-mouse IgG antibody F (ab′)_2_ fragment (Jackson ImmunoResearch Labs 115-065-006), followed by streptavidin-HRP (Jackson ImmunoResearch Labs 016-030-084).

For the evaluation of high avidity IgG, we used a modified ELISA as previously described [22]. Briefly, we included a 10-min incubation with 7 M urea, and we calculated the percent of urea-resistant IgG antibodies by dividing the optical density (OD) of the urea-treated samples by the OD of the untreated samples.

Serum levels of LDH were measured with a commercially available ELISA kit (Antibodies-online #ABIN6957357, Antibodies-Online GmbH: Philadelphia, PA, USA).

### 2.7. Mitostress Test

We used a Mitostress test to measure real-time oxygen consumption rates (OCR) and extracellular acidification rates (ECAR), using a Seahorse XFp extracellular flux analyzer (Agilent, Santa Clara, CA, USA). Briefly, identical numbers of splenic B cells from young and old mice, or of FO and ABCs from young and old mice, were seeded at a concentration of 2.5 × 10^5^/well in a CellTAK (BD Biosciences, Franklin Lakes, NJ, USA)-coated plate. The cells were incubated in an XF DMEM medium supplemented with glutamine, glucose, and pyruvate (200 μL of each reagent in 20 mL of medium), and then exposed to the sequential addition of oligomycin (1 μM), FCCP (fluoro-carbonyl cyanide phenylhydrazone, 5 μM) and rotenone/antimycin (1 μM) to allow for the measure of respiratory parameters.

Seahorse results were confirmed by qPCR experiments in which we measured the gene expression of glucose transporter 1 (Glut1), LDHA (lactate dehydrogenase), and PDHX (pyruvate dehydrogenase), all of which are molecules involved in glucose metabolic pathways.

### 2.8. Statistical Analyses

We used unpaired Student’s *t*-tests (two-tailed) to examine the differences between two groups. We used one-way ANOVA to examine differences between four groups. We used bivariate Pearson’s correlation analyses to examine the relationships between variables. Principal Component Analyses (PCA) were generated using Prism version 10.1.0 software, which was also used to construct all graphs.

## 3. Results

### 3.1. Aging Induces Defects in the Antibody Response to the gp96-Ig-S Vaccine

We immunized young and old mice with the gp96-Ig-S vaccine and evaluated the primary antibody response at different time points after vaccination (t7/t14/t28), as well as the secondary antibody response (t35) after the mice received a boost at t28. The mice were injected intraperitoneally, a route that induces strong systemic and mucosal-specific immunity [21]. Our results show that specific IgG antibodies increase in the serum of the young mice at t7 and t14, remain stable at t28, and increase again after the boost. In the old mice, conversely, specific IgG antibodies increase in the serum at t7, t14, and t28 (at the t14 and t28 time points there is no difference in the amount of IgG between the young and old mice), but not after the boost, clearly indicating an age-associated difference in their response to the vaccine (Figure 1A).

We also measured the avidity of the antibodies generated at t28 and after the boost. We found that the avidity of the antibodies increased from t28 to t35 in young, but not in old, mice. Moreover, we found a reduced avidity in the antibodies at both time points in the old as compared to the young mice, with the difference between the young and old mice being even larger after the boost, emphasizing the importance of measuring the quality in addition to the quantity of the antibodies (Figure 1B). We measured antibody avidity with a modified ELISA method in use in our laboratory, in which urea-resistant antibodies are those of high avidity [22].

### 3.2. Age-Dependent Increase in Serum Levels of the Metabolic Marker Lactate Dehydrogenase (LDH) Are Negatively Associated with gp96-Ig-S Vaccine-Specific Antibodies

Serum samples from the young and old mice were evaluated for the presence of LDH, the enzyme responsible for the conversion of glucose-derived pyruvate into lactate during anaerobic glycolysis. The results in Figure 2A show a significant age-dependent increase in serum levels of LDH at t28 (top) and t35 (bottom). LDH is associated with inflammaging, and therefore considered a risk factor for mortality in patients with serious inflammatory diseases [23,24]. Serum levels of LDH are negatively associated with the amount of gp96-Ig-S vaccine-specific IgG antibodies only at t35 (Figure 2B), and negatively associated with the avidity (Figure 2C) of gp96-Ig-S vaccine-specific IgG antibodies at both time points.

### 3.3. The gp96-Ig-S Vaccine Induces a Metabolic Reprogramming of B Cells from Both Young and Old Mice

The results in Figure 2 suggested to us that metabolic pathways/molecules may be involved in the reduced secondary vaccine response in the old mice. We therefore evaluated the capacity of the gp96-Ig-S vaccine to induce a metabolic reprogramming of B cells from the young and old mice. Using Seahorse technology, we performed a mitostress test of B cells sorted from the spleens of the young and old mice 35 days after vaccination with the gp96-Ig-S vaccine, the time point at which all the mice were sacrificed, and compared them to those from unvaccinated mice. We seeded the B cells into the wells of an extracellular flux analyzer, and we evaluated ECAR (extracellular acidification rates), a measure associated with anaerobic glycolysis and lactate secretion. The results in Figure 3A show a hyper-glycolytic status of the B cells from the old mice (right) as compared to B cells from the young (left) mice. The vaccine decreased the hyper-glycolytic status of B cells from both the young and old mice by significantly decreasing the ECAR, and especially the maximal respiratory capacity (Figure 3B), with better effects being observed in B cells from the old mice. Conversely, the oxygen consumption rates (OCR) measured in the same mitostress test in real time were not affected by the vaccine in both the young and the old mice, as indicated in Figure 3C. These results are consistent with previously published observations that effective vaccine-specific B cell responses are associated with a significant decrease in anaerobic glycolysis and ECAR [25], which is known to support intrinsic inflammation and cell senescence, making the cells refractory/less responsive to further (antigenic/mitogenic) stimulation [26,27,28,29,30].

The mitostress test results were confirmed by qPCR experiments in which we evaluated the RNA expression of *LDHA* (lactate dehydrogenase) and *PDHX* (a component of the pyruvate dehydrogenase complex) that convert glucose-derived pyruvate into lactate or into acetyl-CoA, respectively. LDHA therefore represents a measure of ECAR, while PDHX represents a measure of OCR. Our results show increased levels of RNA transcripts for LDHA in B cells from old versus young mice that were significantly decreased by vaccination only in old mice (Figure 3D). Also, for PDHX, RNA transcripts were higher in B cells from old versus young mice, but no effects of vaccination were observed (Figure 3E).

### 3.4. The gp96-Ig-S Vaccine Decreases RNA Expression of Inflammatory and Senescent Markers in B Cells from Old but not from Young Mice

It is well known from findings obtained in both mice [26] and humans [31] that the metabolic status of an immune cell supports intrinsic inflammation, immune senescence, and pathogenic responses. We wanted to evaluate the intrinsic inflammatory status of B cells from the young and old mice, as well as the effects of the vaccine on the expression of transcripts for inflammatory markers (*TNF, IL6*) and for markers of the senescence-associated secretory phenotype, SASP (*p16^INK4^, p21^CIP1/WAF1^*), measured by qPCR. As expected, based on the results above, we found increased RNA levels of the markers above in B cells from the old mice versus those from the young mice (Figure 4). The results also show a significant decrease in the levels of expression in B cells from the vaccinated mice as compared to the naïve mice.

### 3.5. The gp96-Ig-S Vaccine Decreases the Frequencies of Age-Associated B Cells (ABCs), Which Represent the Most Inflammatory B Cell Subsets, in Old but Not in Young Mice

These increased levels of the expression of markers associated with intrinsic inflammation and senescence are strictly associated with the redistribution of B cell subsets occurring with age (Table 1), leading to the expansion of the subset of age-associated B Cells (ABCs), which represents the most pro-inflammatory B cell subset, at the expense of follicular (FO) B cells, as previously reported [32]. When we measured in the sorted FO and ABCs the mRNA expression of the inflammatory and senescent markers evaluated above, it was found that all of them were higher in the ABCs than in the FO B cells, and more so in the ABCs from the old mice as compared to those from the young mice. These results confirm once again that ABCs are the most pro-inflammatory B cells (Table 2).

The gp96-Ig-S vaccine induced a significant decrease in the frequencies of ABCs in the old but not in the young mice (Figure 5), with the numbers of these cells unchanged as the total number of splenocytes does not change with age. No changes were observed in the frequencies and numbers of MZ B cells. In agreement with the decreased frequencies of ABCs in the old mice after vaccination, the expression of markers of intrinsic inflammation and senescence were found to be decreased in this subset, but not in FO B cells, and only in the old mice (Figure 6).

## 4. Discussion

In this study, we have measured the quantity and quality of antibody responses in young and old mice vaccinated with a vaccine for SARS-CoV-2, and we have shown that the gp96-Ig-S vaccine induces comparable B cell primary responses in young and old mice at later time points, but a significantly lesser secondary response in old as compared to young mice, with the antibodies generated in the secondary response being also of lower avidity. Similar results regarding the effects of aging on vaccine responses have also been shown for other vaccines against respiratory tract infections [33,34,35], suggesting the importance of measuring the quality of the antibody response, i.e., the generation of high-avidity antibodies which are also better functioning. We expect that repeated booster vaccination, as well as increased antigen concentration in the vaccine, will effectively induce higher levels of antibodies which are also of better avidity and neutralizing capacity, as shown during the recent COVID-19 pandemic [36].

Vaccine-induced antibody responses are metabolically regulated, and we show here that gp96-Ig-S vaccine-specific IgG antibodies are negatively associated with serum levels of LDH, a metabolic marker of inflammaging and the risk of mortality in patients with serious inflammatory diseases [23,24]. These age-related increased serum LDH levels reflect increased enzyme activity and lactate secretion. Lactate is an active metabolite in cell signaling with a potent immunoregulatory role, and we have recently demonstrated that lactate induces immunosenescence, intrinsic inflammation, and the secretion of pathogenic autoimmune antibodies in human B cells from elderly individuals [31].

The increased serum LDH suggests a metabolic reprogramming of immune cells involved in the secretion of vaccine-induced specific antibodies. When we investigated the metabolic status of B cells from young and old mice, we found that the gp96-Ig-S vaccine induced a B cell metabolic reprogramming as evaluated by Seahorse and the RNA expression of key metabolic enzymes. In particular, the vaccine decreased the hyper-glycolytic status of B cells from both the young and the old mice, with better effects being observed in B cells from the old mice. This is consistent with previously published observations that effective vaccine-elicited B cell responses are associated with markedly reduced anaerobic glycolysis and ECAR [25], likely because anaerobic glycolysis supports intrinsic inflammation and cell senescence, making the cells refractory/less responsive to further (antigenic/mitogenic) stimulation, as shown in different cell types from both mice and humans [26,27,28,29,30]. Our results showed a decrease in the ECAR, but not in the OCR, profiles of B cells from vaccinated versus naïve mice, clearly showing that metabolic mechanisms are involved, and also suggesting the importance of interventions to modulate the metabolic/inflammatory phenotype of immune cells in order to improve vaccine responses. However, although the gp96-Ig-S vaccine was able to decrease the hyper-glycolytic status of B cells from the old mice, their response was still defective as compared to that of those from young mice, clearly indicating that a higher antigen level and/or repeated booster vaccinations may be needed to improve the response in old mice so that it reaches the level of young mice.

As expected, due to the age-driven expansion of inflammatory B cells, the vaccine-induced decrease in anaerobic glycolysis was associated with a concomitant decrease in the frequencies of the most inflammatory B cells (called ABCs) in the old, but not in young, mice.

In conclusion, in this study we have investigated the causes and mechanisms responsible for the dysfunctional antibody responses that increase with age, using the gp96-Ig-S vaccine with a special focus on immunometabolism. Our results have suggested novel possibilities for therapeutic interventions targeting metabolic pathways with the goal of reducing intrinsic B cell inflammation and improving protective antibody responses. The identification of immunometabolic factors that are correlated with vaccine responses has a translational relevance and may assist in developing targeted interventions to enhance immune responses in vulnerable populations such as elderly individuals with reduced vaccine responses.

## Figures and Tables

**Figure 1 vaccines-12-00960-f001:**
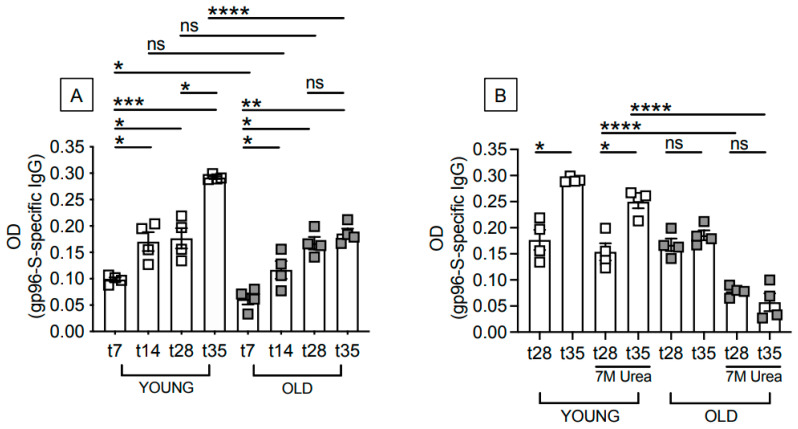
Aging induces defects in the antibody response to the gp96-Ig-S vaccine. Young and old mice were vaccinated with the gp96-Ig-S vaccine at t0, boosted with the same dose at t28 and sacrificed 7 days later. (**A**) Amount of serum antibodies. (**B**) Avidity of serum antibodies at t28 and t35. Urea 7M (final concentration) was added to the samples in the last 10 min before the detection antibody. Samples without 7 M Urea are the same as those shown in (**A**). Statistical analyses were performed with one-way ANOVA. * *p* < 0.05, ** *p* < 0.001, *** *p* < 0.001, **** *p* < 0.0001, ns not significant. Each symbol represents an individual mouse.

**Figure 2 vaccines-12-00960-f002:**
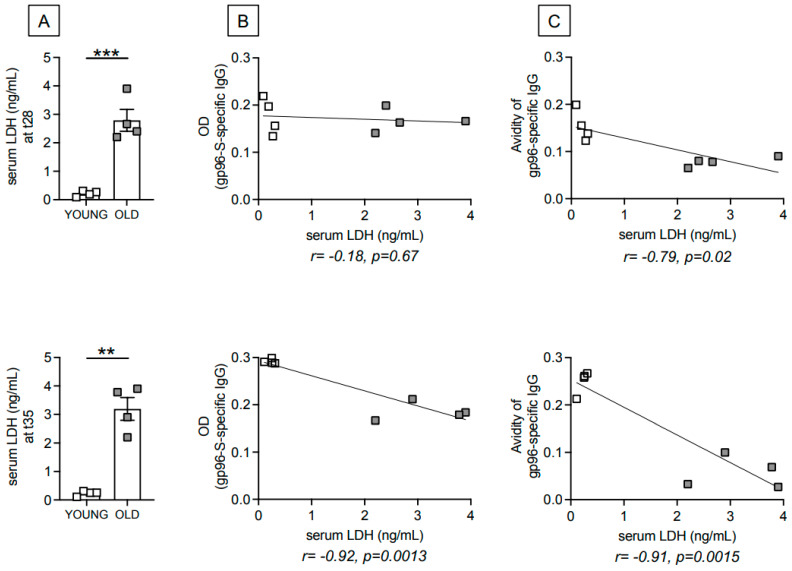
Age-dependent increase in serum levels of the metabolic marker LDH are negatively associated with the avidity of gp96-Ig-S vaccine-specific antibodies. (**A**). Serum levels of LDH were measured by ELISA in the same young and old mice in Figure 1, at t28 (**top**) and t35 (**bottom**) after vaccination. Mean comparisons between groups were performed by unpaired Student’s *t*-test (two-tailed), ** *p* < 0.01, *** *p* < 0.001. Correlation of serum levels of LDH and antibody amounts (**B**) and antibody avidity (**C**) at t28 (**top**) and t35 (**bottom**). Pearson’s correlations and *p* values are shown at the bottom of each graph. Each symbol represents an individual mouse.

**Figure 3 vaccines-12-00960-f003:**
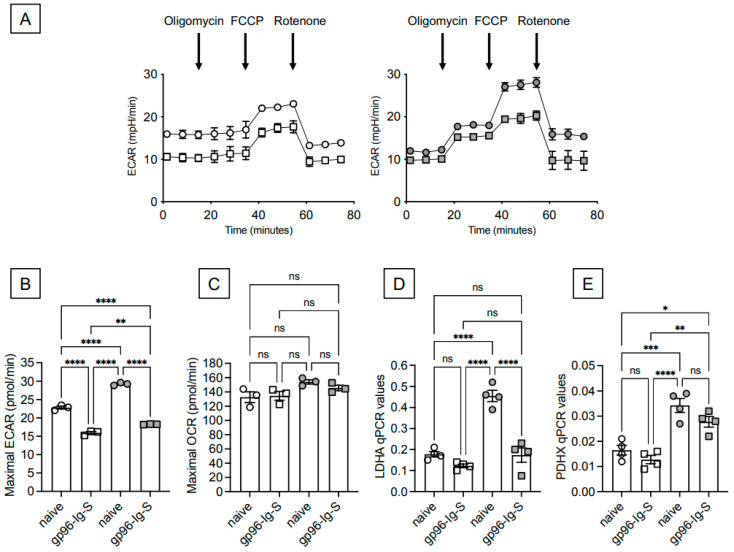
The gp96-Ig-S vaccine induces a metabolic reprogramming of B cells from both young and old mice. B cells were sorted from the spleens of gp96-Ig-S-vaccinated (square symbols) or naïve (round symbols) young mice (white symbols) and old mice (grey symbols). Results are representative of three pairs of young and old mice and are from the t35 time point. After sorting, B cells were left unstimulated and were seeded into the wells of an extracellular flux analyzer at a concentration of 3 × 10^5^/well. (**A**) ECAR results in B cells from young (left) and old (right) mice. (**B**) Maximal ECAR. (**C**) Maximal OCR. (**D**,**E**) After sorting, B cells were left unstimulated. qPCR evaluated LDHA and PDHX mRNA expression, respectively. Results show qPCR values (2^−ΔCt^). Statistical analyses were performed with one-way ANOVA. * *p* < 0.05, ** *p* < 0.01, *** *p* < 0.001, **** *p* < 0.0001, ns not significant. Each symbol represents an individual mouse.

**Figure 4 vaccines-12-00960-f004:**
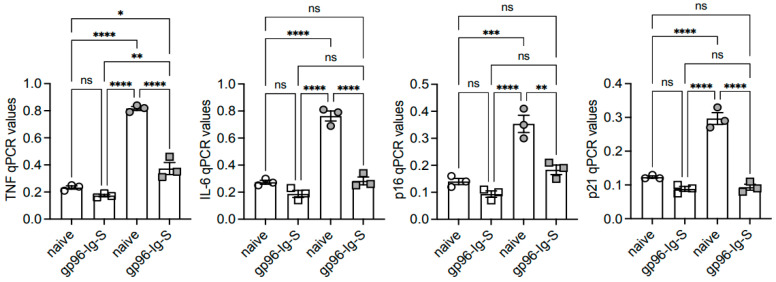
The gp96-Ig-S vaccine decreases RNA expression of inflammatory and senescent markers in B cells from old mice. B cells from the same mice as shown in Figure 3 were left unstimulated, mRNA was extracted and qPCR was run to evaluate mRNA expression of the indicated markers. Results show qPCR values (2^−ΔCt^). Statistical analyses were performed with one-way ANOVA. * *p* < 0.05, ** *p* < 0.01, *** *p* < 0.001, **** *p* < 0.0001, ns not sinificant. Each symbol represents an individual mouse.

**Figure 5 vaccines-12-00960-f005:**
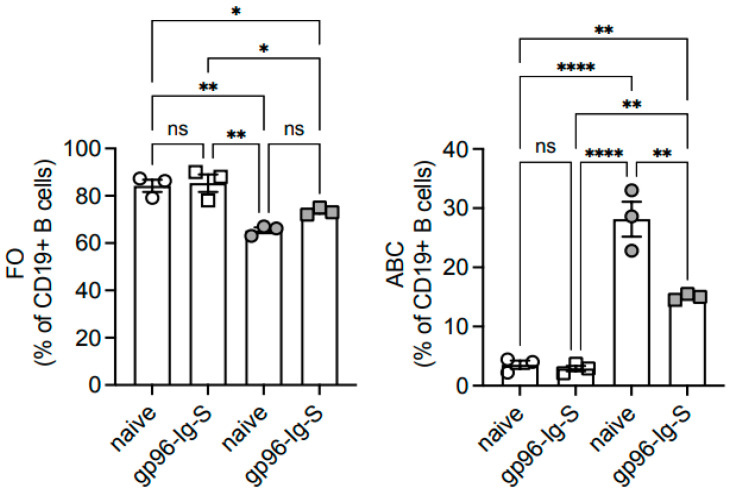
The gp96-Ig-S vaccine decreases the frequencies of ABCs and increases those of FO B Cells in old mice, but not in young mice. Splenic B cells from the same mice shown in Figure 3 were membrane stained to evaluate the frequencies of FO, ABCs, and MZ B cells. Statistical analyses were performed with one-way ANOVA. * *p* < 0.05, ** *p* < 0.01, **** *p* < 0.0001, ns not significant. Each symbol represents an individual mouse.

**Figure 6 vaccines-12-00960-f006:**
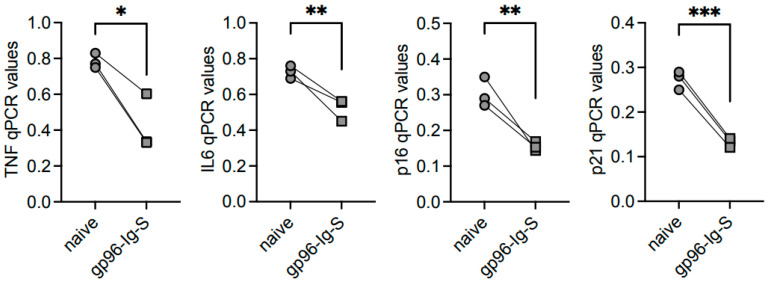
The gp96-Ig-S vaccine decreases RNA expression of inflammatory and senescent markers in ABCs from old mice. ABCs were sorted from the spleens of old mice at t35, the mRNA was extracted and qPCR run to evaluate mRNA expression of the indicated markers. Results show qPCR values (2^−ΔCt^). Statistical analyses were performed with unpaired Student’s *t*-test (two-tailed) to evaluate mean comparisons between groups. * *p* < 0.05, ** *p* < 0.01, *** *p* < 0.001. Each symbol represents an individual mouse.

**Table 1 vaccines-12-00960-t001:** Composition of the splenic B cell pool of young and old naïve mice at t35 time point.

	FO		ABC		MZ	
	Frequencies	Numbers	Frequencies	Numbers	Frequencies	Numbers
YOUNG	84.23 ± 2.59 *	72.32 ± 1.01 **	3.55 ± 0.69 *	3.02 ± 0.51 **	1.30 ± 0.70	1.15 ± 0.51
OLD	65.40 ± 1.22	54.05 ± 0.81	28.13 ± 2.96	23.23 ± 2.29	2.03 ± 0.38	1.69 ± 0.33

The phenotype of FO, ABC, and MZ B cells is shown in Materials and Methods. Results are from three pairs of young and old naïve mice and are from the t35 time point. Statistical analyses were performed with unpaired Student’s *t*-test (two-tailed) to evaluate mean comparisons between groups. * *p* < 0.05, ** *p* < 0.01.

**Table 2 vaccines-12-00960-t002:** Expression of SASP transcripts in B cell subsets sorted from the spleens of young and old naïve mice at t35.

	*TNF*		*IL6*		*p16^INK4^*		*p21^CIP1/WAF1^*	
	FO	ABC	FO	ABC	FO	ABC	FO	ABC
YOUNG	0.004 ± 0.001	0.190 ± 0.012 ####	0.004 ± 0.001	0.024 ± 0.005 ####	0.022 ± 0.019	0.057 ± 0.007 #	0.003 ± 0.002	0.017 ± 0.003 ####
OLD	0.037 ± 0.004 **	0.783 ± 0.024 #### ****	0.024 ± 0.00 *	0.727 ± 0.020 #### ****	0.057 ± 0.007	0.303 ± 0.024 ### **	0.017 ± 0.003 *	0.273 ± 0.012 #### ***

FO and ABCs were sorted from the spleens of young and old naïve mice at t35. Results are from three pairs of young and old mice and are from t35 time point. Mean comparisons between age groups are indicated with asterisk signs, and mean comparisons between subsets are indicated with pound signs; both are calculated by unpaired Student’s *t*-test (two-tailed). # *p* < 0.05, ### *p* < 0.001, #### *p* < 0.0001. * *p* < 0.05, ** *p* < 0.01, *** *p* < 0.001, **** *p* < 0.0001.

## Data Availability

The data presented in this study are available in this article.

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
