# Peer review of "Immunometabolic Regulation of Vaccine-Induced Antibody Responses in Aging Mice"

_vaccines, 2024, doi:10.3390/vaccines12090960_

Round 1

Reviewer 1 Report

Comments and Suggestions for Authors

In this manuscript, the authors evaluated B cell primary and secondary responses and metabolic phenotype or markers in young and old mice after gp96-Ig-S vaccination, revealing that gp96-Ig-S vaccine may reduce the hyper-metabolic status of B cells from old mice. This manuscript is well structured and clearly written. I have the following suggestions that might speed up the publication of this research.

1. Line 42: Why mentioned influenza vaccination specifically?

2. Line 225-229

â‘  Ref 30 is cited inappropriately. Please replace it with other references.

â‘¡ I think relevant experimental results or citations should be provided to elaborate on the point that anaerobic glycolysis supports intrinsic inflammation and senescence of B cells, which would dovetail better with the following results that gp96-Ig-S vaccination decreases RNA expression of inflammatory and senescent markers in B cells.

3. I suggest the authors adjust table 1 and table 2 into the supplementary material.

4. Typo: “Compared” in line 288 and “t23” in line 311

Author Response

Reviewer #1

In this manuscript, the authors evaluated B cell primary and secondary responses and metabolic phenotype or markers in young and old mice after gp96-Ig-S vaccination, revealing that gp96-Ig-S vaccine may reduce the hyper-metabolic status of B cells from old mice. This manuscript is well structured and clearly written.

Thank you!

I have the following suggestions that might speed up the publication of this research.

Line 42: Why mentioned influenza vaccination specifically?

It was a mistake, no need to refer to influenza vaccination. We removed it, thank you!

Line 225-229

â‘  Ref 30 is cited inappropriately. Please replace it with other references.

We apologize with the reviewer but ref. 30 is showing results similar to ours in this manuscript, i.e. effective vaccine-induced B cell responses are associated with a significant reduction in ECAR and anaerobic glycolysis. Therefore, we believe that ref 30 is correct and may remain in the manuscript.

â‘¡ I think relevant experimental results or citations should be provided to elaborate on the point that anaerobic glycolysis supports intrinsic inflammation and senescence of B cells, which would dovetail better with the following results that gp96-Ig-S vaccination decreases RNA expression of inflammatory and senescent markers in B cells.

Thank you for this comment. We agree and we have added citations to support the concept that anaerobic glycolysis supports intrinsic inflammation and cell senescence. Please see paragraph 2.3 “The gp96-Ig-S vaccine induces a metabolic reprogramming of B cells from both young and old mice”.

I suggest the authors adjust table 1 and table 2 into the supplementary material.

No problem to move the tables into Supplementary material.

Typo: “Compared” in line 284 and “t23” in line 311

We corrected both, thank you!

Reviewer 2 Report

Comments and Suggestions for Authors

In the current manuscript, Frasca et al. investigated how metabolic re-programming in immune cells alters humoral immune response against vaccine in aging mice. They demonstrated that compared to young mice, aged mice were severely compromised in mounting secondary immune response against SARS-CoV2 vaccine gp96-Ig-S. They correlated this impairment with increased serum LDH levels in aged mice in which B cells were hyper-glycolytic with increased expression of inflammatory and senescence markers compared to the younger mice. The vaccination reduced this hyper-glycolytic state as well as the inflammatory condition, however, it was still insufficient to rescue the defective humoral response after booster vaccination. The paper provides clear sets of evidence that metabolic reprogramming is involved in the defective antibody response in aging mice. There are only a few minor suggestions for improvement.

1. In Figure 3B-E, the graphs representing the quantification of ECAR, OCR LDH, and PDHX mRNA levels for the young and old mice should be represented together in the same graph (young naïve, vaccinated, old naïve, vaccinated) and should be analyzed by One-way ANOVA instead of t-tests in order to compare the alterations between 4 groups. Although it is clear that there exist differences between young and old, in the current format, a statistical analysis is not possible.

2. The same holds true for the Figures 4-6. The graphs should contain all 4 analysis groups and statistics should be shown for young naïve vs young vaccinated, young naïve vs old naïve, old naïve vs old vaccinated, and young vaccinated vs old vaccinated.

3. In Figure 3D-E as well as in Table 2, the qPCR data should be shown as 2-DDCt (fold change) instead of only DCt.

4. In Figure 1B, there are bars for t28 and t35 without 7M urea. Are they the same as in Fig 1A? if so, the authors should mention in the legend that they represented the same graph once again.

5. There are a few typographical errors in the manuscript that should be carefully checked.

Author Response

Reviewer #2

In the current manuscript, Frasca et al. investigated how metabolic re-programming in immune cells alters humoral immune response against vaccine in aging mice. They demonstrated that compared to young mice, aged mice were severely compromised in mounting secondary immune response against SARS-CoV2 vaccine gp96-Ig-S. They correlated this impairment with increased serum LDH levels in aged mice in which B cells were hyper-glycolytic with increased expression of inflammatory and senescence markers compared to the younger mice. The vaccination reduced this hyper-glycolytic state as well as the inflammatory condition, however, it was still insufficient to rescue the defective humoral response after booster vaccination. The paper provides clear sets of evidence that metabolic reprogramming is involved in the defective antibody response in aging mice.

Thank you very much for your positive feedback!

There are only a few minor suggestions for improvement.

In Figure 3B-E, the graphs representing the quantification of ECAR, OCR LDH, and PDHX mRNA levels for the young and old mice should be represented together in the same graph (young naïve, vaccinated, old naïve, vaccinated) and should be analyzed by One-way ANOVA instead of t-tests in order to compare the alterations between 4 groups. Although it is clear that there exist differences between young and old, in the current format, a statistical analysis is not possible.

Done! Results are in Fig. 3 new and were analyzed by one-way ANOVA. This statistical test has been described in Methods.

The same holds true for the Figures 4-6. The graphs should contain all 4 analysis groups and statistics should be shown for young naïve vs young vaccinated, young naïve vs old naïve, old naïve vs old vaccinated, and young vaccinated vs old vaccinated.

Done as well. We reanalyzed results in Figs. 4 and 5 using ANOVA and we reorganized the presentation of the results. We believe that Fig. 6 may still remain the same as before, it’s different from the other figures (3, 4 and 5), as it is only showing sorted ABCs from old mice. We hope that the reviewer agrees with us.

In Figure 3D-E as well as in Table 2, the qPCR data should be shown as 2-DDCt (fold change) instead of only DCt.

The qPCR values are indeed 2-ΔCts, calculated as follows. First, we calculate the amount of a target gene, relative to GAPDH, and we express this as ΔCt. Then, qPCR values are calculated as 2-ΔCts, not 2-ΔΔCts, as these cells are unstimulated. When they are stimulated, we calculate the ratio of stimulated/unstimulated, which is not the case herein. We added a clarification on these calculations in paragraph 4.5 of Materials and Methods.

In Figure 1B, there are bars for t28 and t35 without 7M urea. Are they the same as in Fig 1A? if so, the authors should mention in the legend that they represented the same graph once again.

The reviewer is correct. We have modified the legend of Fig. 1 explaining that the t28 and t35 bars without 7M Urea are the same in A and B.

There are a few typographical errors in the manuscript that should be carefully checked.

Thank you very much, we checked the paper carefully.
